

# Host plant use drives genetic differentiation in syntopic populations of *Maculinea alcon*

András Tartally[1,2,*], Andreas Kelager[2,3,*], Matthias A. Fürst[2,4] and David R. Nash[2]

[1] Department of Evolutionary Zoology and Human Biology, University of Debrecen, Debrecen, Hungary
[2] Centre for Social Evolution, Department of Biology, University of Copenhagen, Copenhagen, Denmark
[3] Centre for Macroecology, Evolution and Climate, Natural History Museum of Denmark, University of Copenhagen, Copenhagen, Denmark
[4] IST Austria (Institute of Science and Technology Austria), Klosterneuburg, Austria
[*] These authors contributed equally to this work.

## ABSTRACT

The rare socially parasitic butterfly *Maculinea alcon* occurs in two forms, which are characteristic of hygric or xeric habitats and which exploit different host plants and host ants. The status of these two forms has been the subject of considerable controversy. Populations of the two forms are usually spatially distinct, but at Răscruci in Romania both forms occur on the same site (syntopically). We examined the genetic differentiation between the two forms using eight microsatellite markers, and compared with a nearby hygric site, Şardu. Our results showed that while the two forms are strongly differentiated at Răscruci, it is the xeric form there that is most similar to the hygric form at Şardu, and Bayesian clustering algorithms suggest that these two populations have exchanged genes relatively recently. We found strong evidence for population substructuring, caused by high within host ant nest relatedness, indicating very limited dispersal of most ovipositing females, but not association with particular host ant species. Our results are consistent with the results of larger scale phylogeographic studies that suggest that the two forms represent local ecotypes specialising on different host plants, each with a distinct flowering phenology, providing a temporal rather than spatial barrier to gene flow.

# INTRODUCTION

Larvae of *Maculinea* van Eecke (Lepidoptera: Lycaenidae) butterflies start their development on specific host plants. A few weeks later they are adopted into the nests of suitable *Myrmica* Latreille (Hymenoptera: Formicidae) colonies, where they act as social parasites of the ants (*Thomas et al., 1989*). This unusual life cycle has shaped their evolution, as different populations are strongly selected to adapt to different initial host plants and *Myrmica* species depending on their availability (*Thomas et al., 1989*; *Witek et al., 2008*).

Larvae of the *Maculinea alcon* Denis & Schiffermüller group follow a rather specialised development compared to other *Maculinea* species, as they are not simply predators

Corresponding author
David R. Nash, DRNash@bio.ku.dk

of ant brood, but are fed by *Myrmica* workers in preference to their own brood–a behaviour that has been described as a "cuckoo" strategy (*Thomas & Elmes, 1998*). Because they are constantly interacting with worker ants, this means that they need to adapt precisely to the local host ant species, e.g., by mimicking the odour (*Akino et al., 1999*; *Nash et al., 2008*; *Thomas & Settele, 2004*) and sounds (*Barbero et al., 2009*) of the ants, in order to be accepted by a suitable *Myrmica* colony. While the initial host plants of this group are all species of gentian (*Gentiana* L. and *Gentianella* Mönch), they can occur in very different open habitats, such as lowland and mountain meadows or wet and dry swards (*Munguira & Martín, 1999*; *Oostermeijer, Vantveer & Dennijs, 1994*; *Settele, Kühn & Thomas, 2005*; *Tartally, Koschuh & Varga, 2014*). Based on these different habitat types, several forms or (sub)species of the *M. alcon* group have been described. The most widely accepted separation within this group is that the nominotypic *M. alcon* occurs on humid meadows and there is another xerophilous form which has usually been referred to as *M. rebeli* Hirschke (*Thomas et al., 2005*; *Thomas & Settele, 2004*; *Wynhoff, 1998*). Both forms are patchily distributed (*Wynhoff, 1998*) and have been considered as endangered in many European countries (*Munguira & Martín, 1999*), with the xerophilous form considered to be a European endemic (*Munguira & Martín, 1999*). However, several papers (*Habeler, 2008*; *Kudrna & Belicek, 2005*; *Kudrna & Fric, 2013*) have made the case that the xerophilous form is most likely not synonymous with the nominotypic *M. rebeli*, which is found at higher altitude, and has a unique host plant and host ant usage (*Tartally, Koschuh & Varga, 2014*). Furthermore, recent molecular phylogenetic studies (*Als et al., 2004*; *Ugelvig et al., 2011b*; *Bereczki et al., 2015*) suggest that the hygrophilous and xerophilous forms of *M. alcon*, while distinct from other congeners, are not two distinct lineages, and show very little variation in genes normally used for phylogenetic inference. This has been confirmed by several regional population genetic studies (*Bereczki et al., 2005*; *Bereczki, Pecsenye & Varga, 2006*; *Sielezniew et al., 2012*; *Bereczki et al., 2015*), where there is no consistent separation of the two forms. This has led to the current situation where xerophilous and hygrophilous *M. alcon* are not distinguished for conservation purposes, and the species is now considered as "of least concern" in Europe (*Van Swaay et al., 2010*). To avoid confusion, we will refer to the "typical" hygrophilous form of *M. alcon* as '*M. alcon* H' and the xerophilous form as '*M. alcon* X' throughout the rest of this paper, following *Tartally, Koschuh & Varga (2014)*.

The host plant and host ant usage of the two *M. alcon* forms are different, because different gentian and *Myrmica* species are available on the hygric sites of *M. alcon* H and xeric sites of *M. alcon* X. While *M. alcon* H starts development typically on the marsh gentian (*Gentiana pneumonanthe* L.), *M. alcon* X typically uses the cross gentian (*G. cruciata* L.), and there is some evidence that enzyme systems related to host plant use may be diverging in the two forms (*Bereczki et al., 2015*). The development of *M. alcon* X typically continues in nests of *Myrmica schencki* Viereck and *My. sabuleti* Meinert but *M. alcon* H most often uses *My. rubra* L., *My. ruginodis* Nylander or *My. scabrinodis* Nylander as host ants. Furthermore, some other minor or locally important host plant and host ant species have been recorded for both forms (summarized in *Witek, Barbero & Markó, 2014*).

Despite these differences in the host plant and ant usage of *M. alcon H* and *M. alcon X*, phylogenetic reconstruction using morphological and ecological characters suggests that western Palaearctic *M. alcon H* are closer to European *M. alcon X* than Asian *M. alcon H* (*Pech et al., 2004*). In combination, all these results suggest local ecological but not genetic differentiation of the two forms between hygric and xeric sites. Until recently this could only be tested by comparing sites that were separated by tens of kilometres or more, but in the last decade a site has been recorded from Răscruci (Transylvanian basin, Romania) where patches supporting *M. alcon H* and *M. alcon X* occur in a mosaic separated by tens of meters. The two forms use different host plants and mostly different host ants on this site (*Tartally et al., 2008*), and their flying periods are largely separated based on the phenology of their host plants (*Czekes et al., 2014*; *Timuș et al., 2013*). In addition, most previous genetic studies have been based on collecting samples from either flying adults or caterpillars as they emerge from the host plant, which means that any separation by host ant species could not be examined directly. Our aim was therefore to investigate the genetic differentiation between the two forms of *M. alcon* at this unique syntopic site, to relate this to differences in both host plant and host ant use, and to make recommendations for the conservation of the forms based on their shared and predicted future histories (c.f. *Bowen & Roman, 2005*).

## MATERIALS AND METHODS
### Field methods
Two sites in Transylvania (Fig. 1) were visited in the summers of 2007 and 2009 to record host plant and host ant usage and to collect genetic samples of *M. alcon*. Host ant specificity results from 2007 have already been published in *Tartally et al. (2008)*. The first site is at Răscruci (46° 54′N; 23° 47′E; 485 m a.s.l.), which is predominantly an extensively grazed tall-grass meadow steppe with *Gentiana cruciata* (the host plant of *M. alcon X*), but also with numerous small marshy depressions with tall-forb vegetation in which *G. pneumonanthe* (the host plant of *M. alcon H*) is common (*Czekes et al., 2014*). This site gave the unique possibility to compare the host ant specificity and population genetics of *M. alcon H* and *M. alcon X* within the same site. To collect samples, two nearby patches were chosen within this mosaic site where *G. pneumonanthe* and *G. cruciata* were well separated from each other (there was a ca. 20 m wide zone without gentians). In other parts of this site border effects (because of the potential migration of *Myrmica* colonies) or the co-occurrence of the two gentians made it difficult to find *M. alcon* larvae originating clearly from *G. pneumonanthe* or *G. cruciata*. The patch with *G. pneumonanthe* will henceforth be referred to as 'Răscruci wet' (*M. alcon H* patch), while the patch with *G. cruciata* will be referred to as 'Răscruci dry' (*M. alcon X* patch). The nearest known *M. alcon* site (a *M. alcon H* site) to Răscruci is at Şardu (46° 52′N; 23° 24′E; 480 m a.s.l.), 29 km west of Răscruci, which was chosen as a control site. Şardu is a tall-grass, tall-sedge marshy meadow with locally dense stands of *G. pneumonanthe*. The two sites are separated by a range of hills without suitable *M. alcon* habitat (Fig. 1).

To obtain data on the host ant specificity and to get samples for genetic analysis, *Myrmica* nests were searched for within 2 m of randomly selected *Gentiana* host plants, which is

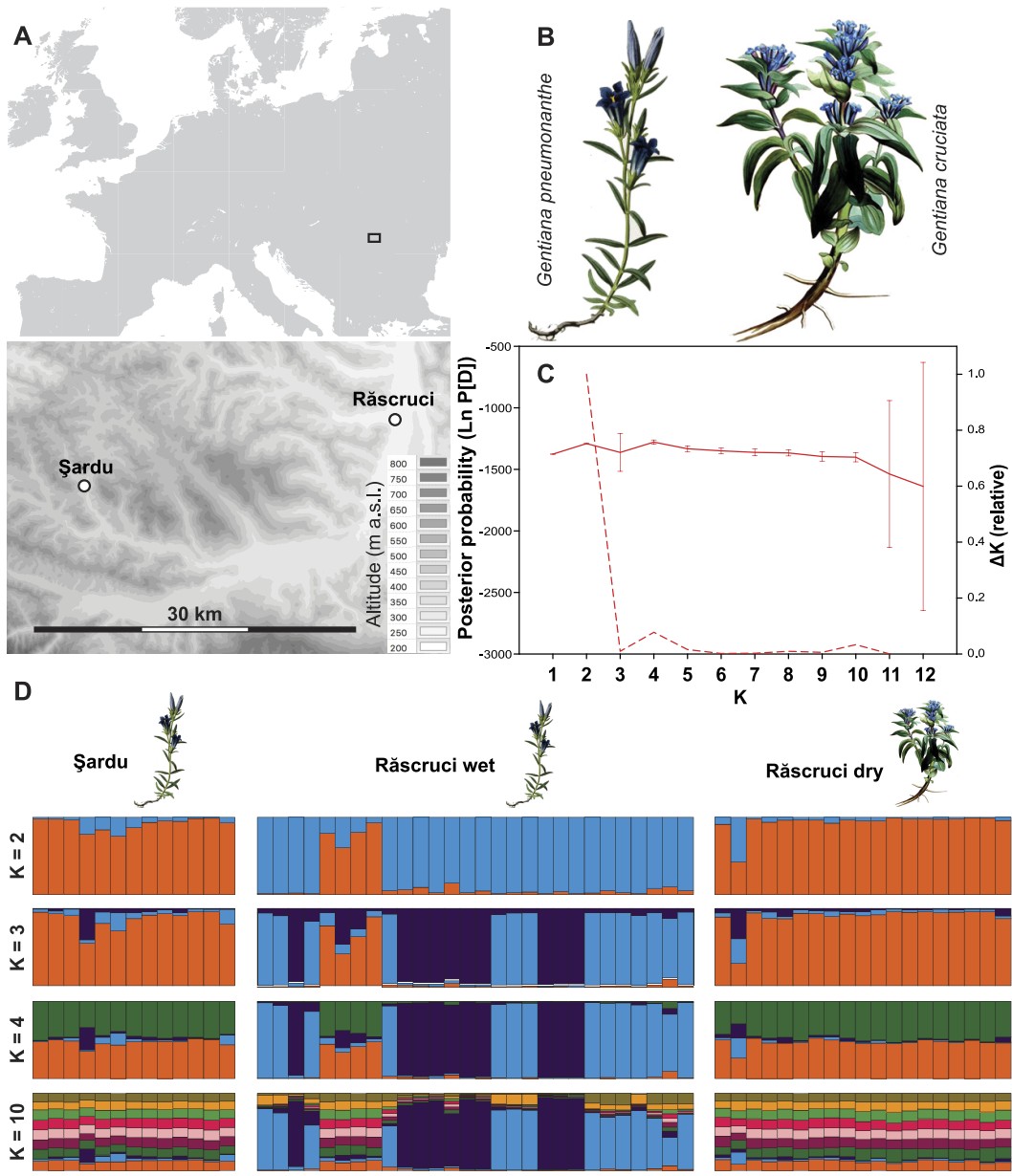

**Figure 1** **Site layout and Bayesian clustering of samples.** (A) Map showing the relative position of the two sample sites. Shaded areas in the detailed map correspond to 50 m contour lines. (B) Initial food plants of the hygric (*G. pneumonanthe*) and xeric (*G. cruciata*) forms of *Maculinea alcon*. Images modified from WikiMedia commons. (C) Posterior probabilities of the number of clusters ($K$) identified by the Bayesian population assignment program Structure. The solid line shows the mean posterior probability for each value of $K$, with error bars representing the standard deviation across simulations. The dashed line shows the $\Delta K$ values of the posterior probabilities from Structure using the method of *Evanno, Regnaut & Goudet (2005)* shown relative to the maximum value of $\Delta K$. Peaks in the value of $\Delta K$ may represent different levels of population substructure. (D) Comparison of genetic clustering of samples into two to four and ten groups using the Bayesian clustering program Structure without a location prior. Each column represents an individual, and is divided according to its probability of membership of cluster 1 (orange), 2 (blue), 3 (dark purple) 4 (green), or 5–10 (other colours).
considered to be the approximate foraging zone of worker ants of the genus *Myrmica* (*Elmes et al., 1998*). Searches were made no earlier than four weeks before the flying period of *M. alcon* at both sites, so that any *M. alcon* caterpillars or pupae found must have survived the winter in the ant nest, and hence have become fully integrated (*Thomas et al., 2005*). Nests were excavated carefully but completely, after which the ground and vegetation were restored to as close to the original conditions as possible. All *M. alcon* caterpillars, pupae and exuviae were counted, placed in 98% ethanol, and stored at $-20$ °C until DNA could be extracted. Five to ten worker ants were also collected from each ant nest and preserved in 70% ethanol for later identification in the laboratory using keys by *Seifert (1988)* and *Radchenko & Elmes (2010)*. For further details, see *Tartally et al. (2008)*.

## Host ant specificity

Host ant specificity (deviation from random occurrence in nests of different *Myrmica* species) was calculated based on the number of fully grown butterfly larvae, pupae and exuviae in two ways: *P1* is the 2-tailed probability from a Fisher exact test of heterogeneity in infection of host ant nests (as implemented at http://www.quantitativeskills.com/sisa/), and *P2* is the probability from a randomization test of ant nests between species, using the software MacSamp (*Tartally et al., 2008*). Published (*Tartally et al., 2008*) and newly-collected data on host ant specificity were combined for these analyses. In the case of Răscruci, host ant specificity results were calculated separately for Răscruci wet and Răscruci dry and also based on the combined data from both patches ('Răscruci both' below).

## DNA extraction and microsatellite analysis

DNA was extracted from approximately 1–2 mm$^3$ of tissue from caterpillars or pupae using a 10% Chelex-10 mM TRIS solution with 5 μl Proteinase K. Samples were incubated at 56 °C for minimum 3.5 h or overnight and boiled at 99.9 °C for 15 min. The supernatant was collected and stored at 5 °C or $-20$ °C for short or long term storage, respectively. For each sample, nine polymorphic nuclear microsatellite loci developed for *Maculinea alcon* were amplified: Macu20, Macu26, Macu28, Macu29, Macu30, Macu31, Macu40, Macu44, and Macu45 (Table 1; *Ugelvig et al., 2011a*; *Ugelvig et al., 2012*) using a REDTaq® ReadyMix™ PCR reaction mix (Sigma-Aldrich). These primer pairs (see concentrations in Table 1) were amplified using standard PCR conditions: initial denaturation for 5 min at 95 °C followed by 30 cycles of 30 s at 95 °C, 30 s at locus-specific annealing temperature (see Table 1) and 30 s extension at 72 °C, finishing with elongation of 15 min at 72 °C run on a Thermo PCR PXE 0.2 Thermal Cycler. Total reaction volume was 10 μl of which 1 μl was template DNA. PCR products were run on a 3130xL Genetic Analyzer with GeneScan 500 LIZ (Life Technologies) as internal size standard and analyzed with GeneMapper® Software version 4.0 (Applied Biosystems). Locus Macu40 could not be scored consistently (excessive stutter bands) and was omitted from all further analysis. The overall proportion of alleles that could not be amplified was 4.6% (see Table S1).

## Tests for Hardy-Weinberg and linkage disequilibrium

The eight microsatellite loci analysed were tested for linkage disequilibrium (genotypic disequilibrium) between all pairs of loci in each sample and for deviations from Hardy–Weinberg proportions using exact tests in FSTAT version 2.9.3.2 (*Goudet, 1995*) based on

**Table 1** Details of microsatellites used in this studyis. Primer concentration in the PCR mix is given below the table.

| Primer | SSR motif | Primer sequences 5′–3′ | SR | $T_A$ | Dye and Multiplex | $N_a$ | Genbank accession | Ref. |
|---|---|---|---|---|---|---|---|---|
| Macu20 | $(CT)_n(AT)_n(CT)_n$ | F: TGGCCCGATTTCCTCTAAAC<br>R: TGCGTGTTTATTTTCATTTTAACAG | 92–122 | 57 | Fam 1 | 9 | HM535963 | U12 |
| Macu26 | $(CA)_n$ | F: CTCCCGGGATAGCATTGAC<br>R: CATTGTCGCGGTCGTAATTC | 92–128 | 57 | Ned 2 | 7 | HM535964 | U12 |
| Macu28 | $(CA)_n(CGCA)_n(CA)_n$ | F: TTTTAATCAAAATCGGTTCATCC<br>R: TCAACCACAAAGCAAGTGAGTC | 195–223 | 57 | Fam2 | 12 | KT851400 | New |
| Macu29 | $(TC)_n$ | F: AAACGCGCTTATGGCTAAAC<br>R: CGGTATGTCCCGTTACATCG | 81–143 | 57 | Vic 3 | 15 | KT851401 | New |
| Macu30 | $(TG)_n$ | F: GACGCGCTGTTATGTATTGC<br>R: CGTCTAGCGTGACCGTAACA | 93–109 | 57 | Pet 4 | 5 | HM586096 | U11 |
| Macu31 | $(GTA)_n(GTC)_n(GTA)_n$ | F: GTTCTGTCCCCCGAACTAGG<br>R: AAACCTGGGATTGGTTAAAAAC | 110–173 | 62 | Ned 5 | 5 | HM586097 | U11 |
| Macu40 | $(CA)_n(GA)_n(CA)_n(GA)_n$<br>$(CA)_n(GA)_n(CA)_n$ | F: CCGTTTGGGAGATACGATGT<br>R: CGCGTGTGCGTATATGTGAT | 110–220 | 57 | Pet 1 | – | KT851402 | New |
| Macu44 | $(AC)_n$ | F: ATAAGTCAGCACGTCAAAGCTG<br>R: TGCAAATACTCCGAATAAATAACTG | 170–220 | 57 | Ned 3 | 10 | HM535965 | U12 |
| Macu45 | $(AC)_n(GC)_n(AC)_n$ | F: TGTGTGACTGCGGTTCTTATC<br>R: TGTAATCGCAGGAGAGATGTG | 145–217 | 57 | Vic 4 | 20 | HM535966 | U12 |

Notes.

SR, product size range (base pairs); $T_A$, Annealing Temperature (°C), Primer dye and Multiplex group; $N_a$, Number of alleles; Ref., Reference source (U11, *Ugelvig et al., 2011a*, U12, *Ugelvig et al., 2012*; New, This study).
Primer concentration in PCR mix: 0.1 ng/μl: Macu20, Macu26, Macu 29, Macu30; 0.2 ng/μl: Macu 28, Macu 31, Macu 40, Macu 44, Macu 45.

480 and 1,260 permutations, respectively. The software package MICRO-CHECKER version 2.2.3 (*Van Oosterhout et al., 2004*) using 1,000 iterations and a Bonferroni corrected 95% confidence interval, was employed to test for possible null-alleles.

## Population structure, genetic differentiation and kinship

We studied the genetic clustering of individual genotypes using the Bayesian algorithm implemented in STRUCTURE version 2.3.4 (*Falush, Stephens & Pritchard, 2003*; *Falush, Stephens & Pritchard, 2007*; *Hubisz et al., 2009*; *Pritchard, Stephens & Donnelly, 2000*). The most likely number of genetically distinct clusters ($K$) was estimated for each $K$ in the range 2–12, allowing for sub-structuring of samples. A burn-in length of 50,000 MCMCs was used to secure approximate statistical stationarity, followed by a simulation run of 500,000 MCMCs using an admixture model with correlated allele frequencies as recommended by *Pritchard, Stephens & Donnelly (2000)*. No location prior was used, and LnP(D) values were averaged over 20 iterations. The most likely value of $K$ (number of clusters) was estimated using the $\Delta K$ method of *Evanno, Regnaut & Goudet (2005)*. To check whether the assumptions inherent in STRUCTURE were biasing our genetic clustering, we also used the Bayesian genetic clustering programs BAPS version 5.2 (*Corander et al., 2008*) and INSTRUCT version 1.0 (*Gao, Williamson & Bustamante, 2007*), which gave essentially identical results (see Additional Analysis S1, Figs. S1 and S2).

For more detailed population differentiation, samples were explored individually as well as in four different partitions: (a) pre-defined populations (**Pop**: Răscruci dry, Răscruci wet and Şardu) which also relates to host plant use (Răscruci wet and Şardu:

*G. pneumonanthe*. Răscruci dry: *G. cruciata*), (b) host ant use (**Ant**: *Myrmica scabrinodis*, *My. sabuleti*, *My. schencki* and *My. vandeli*), (c) host ant nests (**Nest**: specific nest ID within **Pop**), and (d) year of sampling (**Year**: 2007 and 2009), the latter to test for potential temporal differences.

We studied the overall population differentiation between pre-defined populations (**Pop**) calculating *Weir & Cockerham*'s (*1984*) estimate of $F_{ST}$ ($\theta$) using FSTAT version 2.9.3.2 based on 1,000 permutations. As the magnitude of the value of $\theta$ is related to the allelic diversity at the marker loci applied, we further calculated the standardized $G_{ST}''$ (*Meirmans & Hedrick, 2011*) , and the estimator $D_{EST}$ (*Jost, 2008*) as alternative quantifications of genetic differentiation, making comparisons with studies based on other marker loci possible (*Meirmans & Hedrick, 2011*). $G_{ST}''$ and Jost's $D_{EST}$ for pairs of **Pop** samples were calculated using GenoDive version 2.0 b27 (*Meirmans & Van Tienderen, 2004*). Hierarchical AMOVA (Analysis of Molecular Variance: *Excoffier, Smouse & Quattro, 1992*) was calculated for **Pop**, **Ant**, **Nest**, and separately for **Pop** and **Year** using the R-package HierFStat (*Goudet, 2005*) with 9,999 permutations to estimate the variance components and their statistical significance. Individual-based Principal Coordinate Analysis (PCoA) with standardized covariances was employed to obtain a multivariate ordination of individual samples based on pairwise genetic distances, as implemented in the software GenAlEx version 6.502 (*Peakall & Smouse, 2012*). The PCoA were explored for **Nest** within **Ant** within **Pop** across **Year** using nested MANOVA based on the sum of the variances of the different coordinates, as implemented in JMP 12.02 (SAS Institute).

To examine whether the low dispersal ability of *Maculinea alcon* females could lead to high relatedness between samples of individuals collected in the same nest, pairwise measures of kinship (*Loiselle et al., 1995*) and relatedness (*Queller & Goodnight, 1989*) between samples were estimated using GenoDive and GenAlEx respectively. Both of these values estimate the probability that samples share alleles by descent, based on the distribution of alleles in the whole set of samples, with possible values ranging from $-1$ to $+1$. Negative values show that the two individuals compared are less similar in the alleles they share than two randomly picked individuals. Values of kinship and relatedness were then compared between the different partitions of the data using stratified Mantel tests as implemented in GenoDive. To further test the hypothesis that individuals found in the same nest were likely to derive from eggs laid by the same female, the program Colony (v 2.0.6.1; *Jones & Wang, 2010*) was used to give a maximum likelihood estimate of the probability that any two sampled individuals were likely to be either full or half siblings. Since males of *M. alcon* are much more mobile than females, who tend to oviposit in a limited area (*Kőrösi et al., 2008*; DR Nash, 2009, unpublished data),  this was based on a mating system assuming female polygyny and male monogyny, with other parameters kept at their default values.

## RESULTS

### Host ant specificity

A total of 135 *Myrmica* ant nests were found within 2 m of host gentian plants on the two sites, and 90 *Maculinea* larvae, pupae and exuviae were found in 26 infested nests (Table 2)

**Table 2  Details of sampled *Myrmica* nests.** The number of nests found within 2 m of gentians at each site, their infection with *M*. alcon *H* or *M. alcon X*, the number of individual *M. alcon* used for genetic analysis ("Genetic samples": listed in Table S1), and statistical tests of host ant specificity within each site: *P*1, probability from Fisher exact test and *P*2, probability from a randomization test of ant nests between species. Significant *P*-values ($P < 0.05$) are marked in bold.

| Site | Maculinea | Myrmica | No. nests | No. with *M. alcon* | P1 | No. of *M. alcon* | Range | P2 | Genetic samples |
|------|-----------|---------|-----------|---------------------|----|-------------------|-------|----|-----------------|
| Răscruci dry | alcon X | sabuleti | 10 | 5 | **0.004** | 17 | 1–8 | **0.002** | 13 |
| | | schencki | 6 | 2 | | 18 | 1–15 | | 5 |
| | | scabrinodis | 23 | 1 | | 1 | | | 1 |
| Răscruci wet | alcon H | scabrinodis | 31 | 9 | – | 30 | 1–7 | – | 28 |
| Răscruci both | both | as above | | | 0.078 | | | **0.021** | |
| Șardu | alcon H | vandeli | 27 | 2 | 0.147 | 9 | 2–7 | 0.495 | 2 |
| | | scabrinodis | 38 | 7 | | 15 | 1–4 | | 11 |

including 87 nests and 56 *Maculinea* already published in *Tartally et al. (2008)*. Altogether four *Myrmica* species were found. Only *My. scabrinodis* was present at all sites, and was the most abundant ant species (59% of all ant colonies found). This species was used as a host on all three sites. Only a single *M. alcon X* was found in a nest of *My. scabrinodis* at Răscruci dry despite the dominance of this ant there and its frequent usage by *M. alcon H* on Răscruci wet (Fisher's exact test, $P = 0.032$). The much greater exploitation rates of *My. sabuleti* and *My. schencki* led to significant overall host ant specificity at Răscruci dry (Table 2).

## Genetic diversity and inbreeding

Measures of genetic diversity and *F*-statistics generated by FSTAT for each locus are listed in Table S2 in the supporting information. Analysis with Micro-Checker revealed that Macu29 had a highly significant ($P < 0.001$) excess of homozygotes and cases of non-amplification consistent with the presence of a relatively high proportion (>20%) of null-alleles, and was therefore excluded from further analysis. All other loci showed no significant deviations from Hardy-Weinberg proportions. Tests for linkage disequilibrium revealed only a few sporadic significant results showing no overall pattern (Table S3), so all loci were retained in further analysis, which was thus based on seven polymorphic loci. All three of the pre-defined populations showed no evidence of inbreeding (Table 3), and in fact showed negative values for the inbreeding coefficient $F_{IS}$ (meaning an excess of heterozygotes), although not significantly so (Table 3).

## Population structure

Structure analysis revealed rather invariable log-likelihood values for partitioning of the data into genetic clusters, but the highest change in log-probability value was for $K = 2$, with lower maxima at $K = 4$ and $K = 10$ (Fig. 1). There was a clear overall distinction between samples from Răscruci wet in one genetic group and Răscruci dry and Șardu in another group. Levels of admixture between genetic clusters were generally low, but four individuals from Răscruci wet (from two different *My. scabrinodis* nests) showed high affinity to the Răscruci dry-Șardu group, irrespective of the value of *K*. One individual found in a *My. sabuleti* nest at Răscruci dry (sample code: DA14) appeared genetically more

**Table 3  Pairwise differentiation between, and inbreeding and genetic diversity within predefined populations.** Values above the diagonal in the matrix (with blue background) are $\theta$ ($F_{ST}$), values along the diagonal (with green background) are $F_{IS}$, values below the diagonal (with yellow background) are $G''_{ST}/D_{EST}$. Values in bold differ significantly from zero ($P < 0.001$). Below the matrix are mean values ($\pm$SE) of four different measures of within-population genetic diversity. The effective number of alleles per locus ($A_E$), the observed heterozygosity ($H_O$), the expected heterozygosity ($H_E$) and the unbiased expected heterozygosity ($uH_E$). $P$-values for comparisons between pre-defined populations based on mixed model comparison across loci are shown on the right.

| | Răscruci dry | Răscruci wet | Șardu | $P$ |
|---|---|---|---|---|
| Răscruci dry | −0.050 | **0.093** | **0.059** | |
| Răscruci wet | **0.302/0.255** | −0.106 | **0.103** | |
| Șardu | **0.207/0.151** | **0.330/0.221** | −0.052 | |
| $A_E$ | 4.000 ± 0.606 | 3.365 ± 0.418 | 4.594 ± 0.909 | 0.131 |
| $H_O$ | 0.707 ± 0.067 | 0.717 ± 0.056 | 0.733 ± 0.090 | 0.944 |
| $H_E$ | 0.707 ± 0.046 | 0.674 ± 0.034 | 0.709 ± 0.063 | 0.781 |
| $uH_E$ | 0.729 ± 0.048 | 0.687 ± 0.035 | 0.738 ± 0.066 | 0.613 |

similar to those from Răscruci wet. For values of $K$ higher than 2 there was no additional partitioning between the pre-defined populations, but some substructure in Răscruci wet became apparent for $K = 3$, with two partitions that were relatively dissimilar, while for $K \geq 4$ no additional grouping of individuals was apparent (Fig. 1).

## Genetic differentiation

We found significant overall genetic differentiation between pre-defined populations ($\theta = 0.090$, $D_{EST} = 0.215$; Table S2). Pairwise genetic differentiation measures $\theta$, $G''_{ST}$ and $D_{EST}$ were significant for all population comparisons after Bonferroni adjustment ($P < 0.003$; Table 3). There was no evidence of inbreeding, either overall ($F_{IS} = -0.074$, Table S2), or within pre-defined populations (Table 3).

Hierarchical AMOVA (Table 4) revealed that most genetic variance (93.4%) was within individuals, but that significant variation was also explained by **Pop** and **Nest**. The proportion of variation and inbreeding coefficients for individuals within **Nest** and **Ant** were both negative, indicating that there was greater heterozygosity between individuals in the same nest and between samples across different *Myrmica* species than between randomly chosen samples from the data set. Samples from different years explained only 0.12% of the genetic variance in a separate AMOVA ($P = 0.354$). The Principal Coordinate Analysis retained a total of six principal coordinates with eigenvalues greater than 1, which together explained 52% of the variance in genetic distance (13.5%, 10.3%, 8.4%, 8.0%, 7.2% and 4.5% for coordinates 1–6 respectively; see Fig. S3 for more details). These showed a similar result to the AMOVA where **Year** samples (2007 and 2009) overlapped completely in genetic ordination space ($F_{1,36} = 8.91 \times 10^{-16}$, $P = 0.999$), while samples from Răscruci wet were separated from those from Răscruci dry and Șardu (Fig. 2; $F_{2,36} = 6.08$, $P = 0.005$). We found a pronounced structuring of samples when examining nests within pre-defined populations ($F_{18,36} = 3.59$, $P < 0.001$), with samples from the same nest clustering together, but there was no consistent clustering of samples from the nests of the same host ant species

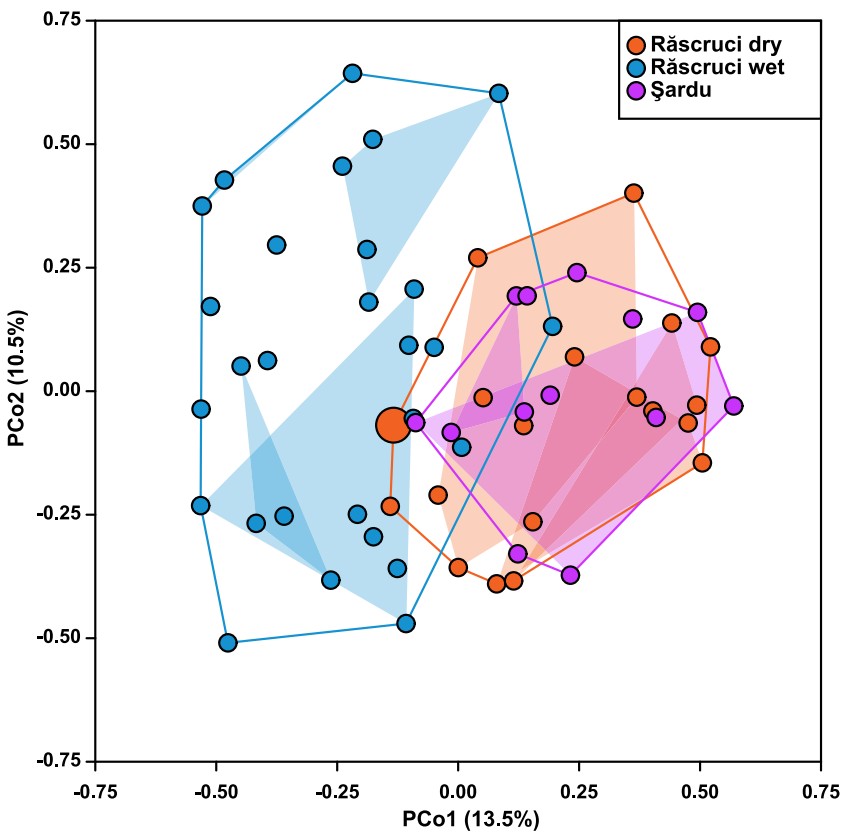

**Figure 2** **Ordination of samples based on principal coordinate analysis.** Each symbol represents an individual, coloured according to its pre-defined population (blue, Răscruci wet, orange, Răscruci dry, purple, Şardu). Coloured lines are convex hulls enclosing all samples from each pre-defined population, while coloured regions are convex hulls enclosing samples collected from the same nest. The single individual (sample DB15) collected from a *My. scabrinodis* nest at Răscruci dry is shown with a larger symbol.

**Table 4** **Hierarchical analysis of molecular variance.** Calculated using HierFStat (*Goudet, 2005*). The *F*-coefficient gives the estimated inbreeding coefficient (excess of homozygotes) at each hierarchical level. *P*-values are based on 1,000 re-samplings of the data.

| Source | d.f. | Variance component | %variance | *F*-coefficient | *P* |
|---|---|---|---|---|---|
| Between **Pop** | 2 | 0.577 | 10.6 | 0.106 | 0.027 |
| **Ant** within **Pop** | 3 | −0.257 | −4.7 | −0.053 | 0.956 |
| **Nest** within **Ant** | 18 | 0.609 | 11.2 | 0.119 | <0.001 |
| Individuals within **Nest** | 36 | −0.569 | −10.5 | −0.126 | >0.999 |
| Within Individuals | 60 | 5.067 | 93.4 | | |
| Total | 119 | 5.427 | | | |

($F_{3,36} = 0.887$, $P = 0.457$). The single sample from Răscruci dry that was collected from a *My. scabrinodis* nest (sample code DB15) had a first principal component that was more characteristic of samples from Răscruci wet (which all used this host ant species; Fig. 2), but was not assigned to this population in the Bayesian analysis (Fig. 1).

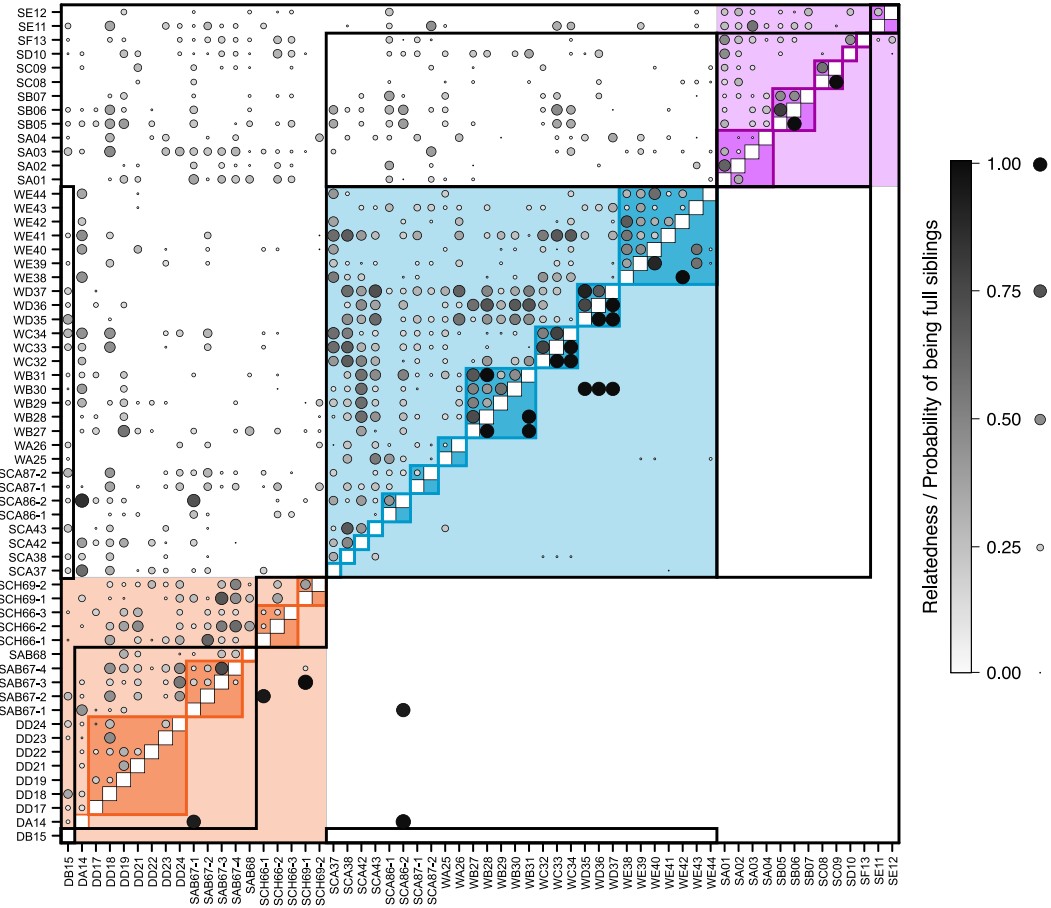

**Figure 3  Relatedness and parentage analysis of samples.** The pairwise matrix shows the estimated *Queller & Goodnight (1989)* relatedness of each pair of individuals (excluding those with negative relatedness) above the diagonal, and the probability that each pair are full siblings based on maximum likelihood estimates from Colony (*Jones & Wang, 2010*) below the diagonal. Comparisons between samples from the same pre-defined population are shaded according to the same colour scheme as Fig. 2 (blue, Răscruci wet, orange, Răscruci dry, purple, Șardu). Individuals sharing the same ant nest are outlined with lines in these same colours, and those sharing the same *Myrmica* species as host are outlined with black lines. The area and shade of each data point is proportional to the relatedness or probability of being full siblings for that pair of individuals.

## Kinship and parentage analysis

Overall pairwise relatedness of individuals (Fig. 3) sampled from the same nest (0.311) was significantly higher than that of those sampled from different nests within the same site (−0.032; Stratified Mantel test: $r^2 = 0.069$, $P \leq 0.0001$). Looking at individual sites, the same pattern was found at both *M. alcon H* sites (Răscruci wet: within-nest relatedness = 0.32, between nests = 0.10, $r^2 = 0.130$, $P = 0.0002$; Șardu: within-nest = 0.249, between nests = 0.065, $r^2 = 0.126$, $P = 0.002$), but relatedness was not significantly different within and between nests at Răscruci dry (within-nest = 0.003, between nests = 0.081, $r^2 = 0.001$, $P = 0.448$). Similar results were found when analysing pairwise kinship (see Fig. S4 and Additional Analysis S2). Maximum-likelihood analysis using COLONY identified 38 pairs of individuals as potential full siblings (with probabilities ranging from 0.002 to 0.935; Fig. 3),

and 161 as potential half siblings (with probabilities ranging from 0.002 to 0.742; Fig. S4). For the 21 pairs with high (>0.5) probability of being full siblings, 13 (62%) were from the same nest, six were from nests of the same ant species at the same site, and only two were from different sites, both including individual SCA86-2 from Răscruci wet, which appears closely related to two individuals (DA14 and SAB67-1) from different *Myrmica* nests from Răscruci dry (Fig. 3). This last result probably reflects the non-amplification of characteristic loci for these individuals (see Table S1), so that they share common alleles without being related. Within sites, a high proportion of individuals from both Răscruci wet (60.4%) and Şardu (37.5%) from multiply-infested nest had individuals estimated to be full siblings in the same nest (overall 52.9%, Generalized linear model with binomial errors and Firth corrected maximum likelihood, comparing sites: Likelihood-ratio $\chi^2 = 0.73$ $d.f. = 1$, $P = 0.391$). However, none of the *M. alcon X* individuals sharing nests at Răscruci dry were estimated to be full siblings (*M. alcon X* vs. *M. alcon H*: likelihood-ratio $\chi^2 = 9.72$, $d.f. = 1$, $P = 0.002$).

Although there was evidence for strong within-nest relatedness of individuals, the patterns of genetic diversity and differentiation were not strongly affected by this, and were unaffected when analyses were repeated with only a single individual from each nest (see Table S4).

## DISCUSSION

This study gives the first comparison of the host ant specificity and genetic composition of *M. alcon H* and *M. alcon X* within the same site.

The host ant specificity found in this study confirms the earlier results of *Tartally et al. (2008)* that these populations use the typical host ants found in other Central European studies (*Höttinger, Schlick-Steiner & Steiner, 2003*; *Sielezniew & Stankiewicz, 2004*; *Steiner et al., 2003*; *Tartally et al., 2008*; *Witek et al., 2008*). *M. alcon H* was found exclusively with *My. scabrinodis* at Răscruci wet and also with *My. vandeli* at Şardu, but *M. alcon X* was found mainly with *My. sabuleti* and *My. schencki* at Răscruci dry. Interestingly only one *M. alcon X* was found with *My. scabrinodis*, despite this *Myrmica* species being the most numerous at Răscruci dry (Table 2) and being the main host of *M. alcon X* in two other sites in the Carpathian-Basin (*Tartally et al., 2008*). *My. scabrinodis* usage could therefore be a potential link between the *M. alcon H* and *M. alcon X* populations at Răscruci (and probably in some other regions), but *M. alcon X* shows a clear separation from the *M. alcon H* in the proportional usage of this host ant. The background of this separation in the host ant specificity of *M. alcon H* and *M. alcon X* at Răscruci is not clear, but could reflect the dynamic arms race between the different genetic lineages of *M. alcon* and local host ants (*Nash et al., 2008*).

Our genetic results (Figs. 1 and 2) show strong genetic differentiation between *M. alcon H* and *M. alcon X* at Răscruci, indicating limited gene flow between these two groups, although it is interesting to note that a few individuals had genotypes more characteristic of the other population. This differentiation is likely due to separation in time rather than space because of the different phenology of the host plants, which results in largely

non-overlapping flying seasons of *M. alcon H* and *M. alcon X* (*Timuș et al., 2013*). This may be reinforced by lowered fitness of any hybrid individuals that would emerge during the approximately 2-week gap when neither host plant is suitable for oviposition.

The lowest level of between-population differentiation, on the other hand, was between *M. alcon X* from Răscruci and *M. alcon H* from Șardu, and Bayesian population assignment suggests that these are so similar that they have almost certainly been part of a single population. This supports previous findings of no overall phylogenetic differentiation between the two forms of *M. alcon* (*Als et al., 2004*; *Fric et al., 2007*; *Ugelvig et al., 2011b*), and that the two forms tend to be more genetically similar regionally than either is to more distant populations that use the same host plant (*Bereczki et al., 2005*; *Bereczki, Pecsenye & Varga, 2006*; *Pecsenye et al., 2007*). Hence the two forms cannot be regarded as host races (*Drès & Mallet, 2002*), since they do not fulfill the criterion of spatial and temporal replicability. Genetic analysis of several Polish and Lithuanian *M. alcon* populations using microsatellite markers (*Sielezniew et al., 2012*) gave similar results to ours (Figs. 1 and 2) in that there was no clear pattern reflecting genetic division into two ecotypes. Sielezniew et al. also found that the *M. alcon X* ecotype was less polymorphic, and its populations more differentiated than those of the *M. alcon H* ecotype. Their data also suggest that *M. alcon H* populations form a single clade but *M. alcon X* can be split into more clades, suggesting that *M. alcon H* is an ancestral form and that *M. alcon X* represents a group of independently evolved *M. alcon H* populations that have switched to use dryer habitats with the locally available *Gentiana* and *Myrmica* species. They propose that the background of this pattern may be independent specialisations on different host ant species, since in their study clades of *M. alcon X* largely reflected host ant use. However, we find no evidence of genetic differentiation associated with host-ant usage at Răscruci or Șardu (Figs. 1– 3), and no difference in genetic diversity in populations of the two ecotypes (Table 3). Due to the relatively large distances and potential barriers between Răscruci and Șardu it is unlikely that there has been recent gene-flow between the two sites, which suggests that Răscruci was likely colonized at least twice from two different gene pools, and that the ancestors of the Răscruci wet population are no longer locally extant (or have evaded detection).

The higher within-nest than between-nest relatedness between individuals of *M. alcon H* is consistent with observations of limited dispersal of ovipositing females (*Kőrösi et al., 2008*) which is likely to lead to substantial within-population substructure between nests, as found here. The additional grouping of individuals from $K = 2$ to $K = 4$ in our population assignment analysis groups families of relatives within populations. Our parentage analysis confirms the relatively high probability that *M. alcon H* caterpillars infesting the same *Myrmica* nest are relatives, and in around 50% of cases may be full siblings. The lack of any association between relatedness and infestation of nests for *M. alcon X* from the Răscruci dry site is consistent with the difference in oviposition strategy and mobility of butterflies from this population compared with those from the Răscruci wet site (*Czekes et al., 2014*; *Timuș et al., 2013*).

Given the small size of these *M. alcon* populations (*Timuș et al., 2013*) and their low dispersal (*Kőrösi et al., 2008*), it is interesting to note that there is no evidence of inbreeding

among the individuals that we examined, and that estimated inbreeding coefficients ($F_{IS}$ and $G_{IS}$) were negative. This means that individuals were more heterozygous than expected if mating were random (albeit not significantly so), which is probably a result of difference in mating strategy of males and females. Males are highly mobile, and tend to patrol a large area while seeking females, whereas females are rather sedentary, and are often mated immediately after emerging from the *Myrmica* nest in which they developed, and then go on to lay eggs on host plants relatively close by (*Kőrösi et al., 2008*). This means that females are unlikely to be related to their mates, and may in fact be less related to them than to a randomly chosen male, because males pupate and emerge several days before females (*Meyer-Hozak, 2000*). This, together with the observation that caterpillars that develop in the same *Myrmica* nest may be offspring of the same female, can easily lead to the negative inbreeding indices observed both overall, and for individuals within *Myrmica* nests and host ant species within pre-defined populations within our hierarchical AMOVA.

Regardless of its origin, it is clear that the *M. alcon X* population at Răscruci is ecologically highly differentiated from the local *M. alcon H* populations in terms of its host plant and host ant use, as well as in its behaviour (*Czekes et al., 2014*; *Timuș et al., 2013*). The pattern of differentiation we see is not that typically associated with speciation via host race formation (*Drès & Mallet, 2002*), since the two forms do not fulfill the criterion of spatial and temporal replicability (i.e., "*are more genetically differentiated from populations on another host in sympatry (and at the same time) than at least some geographically distant populations on the same host*"; *Drès & Mallet, 2002*, p. 473–4). *M. alcon X* cannot, therefore, represent an evolutionarily significant unit in conservation terms as usually defined, but we would argue that it should still be regarded as a functional conservation unit (*Casacci, Barbero & Balletto, 2013*). The site at Răscruci represents the only known area where both forms of *M. alcon* occur syntopically, and so is of particular value to research on speciation, and has great potential for examining adaptation at non-neutral genetic markers. This is enhanced by the occurrence of two other *Maculinea* species on the same site; *M. teleius* Bergsträsser (*Tartally & Varga, 2008*) and *M. nausithous kijevensis* Sheljuzhko (*Rákosy et al., 2010*; *Tartally & Varga, 2008*), as well as the *Myrmica* parasites *Microdon myrmicae* Schönrogge et al. (Diptera: Syrphidae; *Bonelli et al., 2011*) and *Rickia wasmannii* Cavara (Ascomycota: Laboulbeniales; *Tartally, Szűcs & Ebsen, 2007*). The *Maculinea* spp. parasitoid *Ichneumon eumerus* Wesmael (Hymenoptera: Ichneumonidae) is also present (*Tartally, 2008*). Most of these species are also found at Şardu (except *M. alcon X* and *M. nausithous*: *Tartally, 2008*). It should be emphasized that all of these species can be found in the nests of, and ultimately depend on, *My. scabrinodis* (as well as other *Myrmica* species: see *Witek, Barbero & Markó (2014)* for a review), providing a unique opportunity to examine a complex set of parasitic interactions revolving around a single keystone ant species.

## CONCLUSION

Our analysis of *Maculinea alcon* from a unique site where both the xerophilous and hygrophilous forms of this butterfly are found within tens of meters of each other has demonstrated strong genetic differentiation between the two forms. However,

the xerophilous form was not significantly differentiated from the next closest known population of the hygrophilous form. This supports other recent work suggesting that the hygrophilous and xerophilous forms are not separate species or even subspecies, and that the name *M. rebeli* has frequently been applied to the xerophilous form incorrectly. There is some overlap in host ant species use between the two forms, so the most likely proximate reason for the local genetic differentiation found is differences in host plant phenology. We suggest that since the two different forms of *M. alcon* do not have a separate evolutionary history, they cannot be regarded as "evolutionarily significant units" for conservation as the term was originally used (*Bowen & Roman, 2005*; *Casacci, Barbero & Balletto, 2013*). However, since they represent current ecological diversification of the group, and may have different evolutionary potentials (e.g., through selection on different enzyme systems; *Bereczki et al., 2015*), they should continue to be treated as separate management units for long-term conservation (*Bowen & Roman, 2005*; *Casacci, Barbero & Balletto, 2013*). Hence, we support the continued separation of the two forms in future studies to further explore their evolutionary trajectories and conservation potential.

## ACKNOWLEDGEMENTS

We thank Tibor-Csaba Vizauer, László Rákosy and Zoltán Varga for assistance in the field and Shukriya Barzinci, Maria Mikkelsen and Sylvia Mathiasen for assistance in the laboratory. Zoltán Varga, Enikő Tóth, Simona Bonelli and Robert Toonen provided valuable comments on earlier versions of the manuscript.

### Funding

AT was supported by a Marie Curie Intra-European Fellowship and a Marie Curie Career Integration Grant within the 7th European Community Framework Programme, and by a 'Bolyai János' scholarship of the Hungarian Academy of Sciences (MTA). AK and DRN were supported by a Danish National Research Foundation grant to the Centre for Social Evolution (DNRF57) and the Center for Macroecology, Evolution and Climate. The funders had no role in study design, data collection and analysis, decision to publish, or preparation of the manuscript.

### Grant Disclosures

The following grant information was disclosed by the authors:
Marie Curie Intra-European Fellowship.
Marie Curie Career Integration.
Bolyai János scholarship.
Danish National Research Foundation: DNRF57.
Center for Macroecology, Evolution and Climate.

### Competing Interests

The authors declare there are no competing interests.

## Author Contributions

- András Tartally conceived and designed the experiments, performed the experiments, analyzed the data, wrote the paper, reviewed drafts of the paper.
- Andreas Kelager performed the experiments, analyzed the data, wrote the paper, prepared figures and/or tables, reviewed drafts of the paper.
- Matthias A. Fürst performed the experiments, analyzed the data, wrote the paper, reviewed drafts of the paper.
- David R. Nash conceived and designed the experiments, analyzed the data, wrote the paper, prepared figures and/or tables, reviewed drafts of the paper.

## DNA Deposition

The following information was supplied regarding the deposition of DNA sequences:
GenBank accession numbers included in Table 1.

## Data Availability

The raw data has been supplied as a Supplemental Dataset.

## Supplemental Information

Supplemental information for this article can be found online at http://dx.doi.org/10.7717/peerj.1865#supplemental-information.

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
