# Peer review of "Host plant use drives genetic differentiation in syntopic populations of Maculinea alcon"

_PeerJ, doi:10.7717/peerj.1865_

## Round 0.1 · original submission · Major Revisions

I am sorry for the delay in getting your manuscript back to you. The second referee asked for several extensions to complete their review, but then resigned and left us with only a single review. Rather than delay your manuscript further by requesting a new review, I have reviewed the manuscript myself as the second referee and provide you with my comments below. Most of the suggestions from us both are relatively minor, but that will improve the manuscript prior to publication. I have listed this as a major revision based on the request for additional of kinship analyses, but I leave it up to the author as to whether or not you want to actually include these additional analyses. I think it would be valuable to add to the manuscript, and I am curious about the results, but I do not expect that it will change anything about your results or your conclusions drawn from the study, so I leave it to you to decide if you want to add them or not.

In addition to the suggestion of the first referee to shorten and simplify the text, I also have a number of suggestions for you that I believe will make the manuscript easier to read if you decide to make the changes. For example, although there is nothing wrong with the way that you have presented your analyses, I would argue that you could simplify the entire genetic clustering section and conserve space by just selecting one analysis for the paper. I understand why you have used all 3 methods, but they are all giving us essentially the same information, so I would suggest removing 2 of them from the main body of the text and either moving the other 2 to supplementary materials where the K analyses are now. If you present one in detail and just say that the other 2 gave essentially the same result would convey the same information in a more concise way. Instead of using the room to describe 3 different analyses with the same outcome, I would like to see more explicit discussion of your ΔK analyses for the option you select in the body of the manuscript rather than the supplement. Personally, I would rather that you included one analysis (say Structure) with K=2 vs K=3 vs K=4 vs K=10 so that I can see for myself how well each matches the sampling scheme.

I liked the relatedness analysis, but would also have liked to see a kinship analysis to complement that. There are a number of good software options out there now for kinship analyses, and they provide great insights in some systems (e.g., Iacchei, et al. 2013. Mol Ecol, 22:3476-3494.) In this case it would allow us to determine whether your increased relatedness is driven by overall inbreeding among isolated subpopulations or a few highly successful families which result in sampling of siblings. You do not present inbreeding coefficients in the manuscript other than the InStruct analysis but with Fis values approaching 0.5 as you move from sites to individual nests with increasing K, it seems possible that you do have kin structure within nests, but it would be nice to know the degree of inbreeding and the degree of kin structure among your samples.

I was glad to see your inclusion of standardized FST metrics in the analyses in the section starting on line 200, but feel that this is an area of some controversy still, so I would ask that you point unfamiliar readers to at least one of the many reviews of the issue. I can suggest a couple of my favorites if you do not have one of your own. 1) Meirmans, P. G., & Hedrick, P. W. (2011). Assessing population structure: FST and related measures. Molecular Ecology Resources, 11:5-18. 2) Bird, C. E., Karl, S. A., Smouse, P. E., & Toonen, R. J. (2011). Detecting and measuring genetic differentiation. Phylogeography and Population Genetics in Crustacea, 19:31-55. I would also have liked to see the actual AMOVA table presented so that the reader can see your design and the values for each of the variance components. It is fine to describe the results in the text, but the table is the actual data, and I would like to see it included either in the text of the supplementary files. Likewise, you describe pairwise FST values in the text, but never present them to the reader – I would like to see the full pairwise FST table included somewhere in the revision (perhaps the supplementary file). Finally, it was unclear to me why you switched among the different software packages given that most of your statistics could be calculated within either one of them. It is certainly not a major point, but I found it unusual and other readers may ponder this as well. If there was a specific reason for this, you may want to explain it here.

For someone who is not familiar with your system or study site, it would be really useful to provide some map of the area so that there is a better sense of the differences. Along these same lines, you give us GPS but never simply tell the reader exactly how far apart are the sites, which would be useful to know.

Lastly, based on the data presented here, you don't yet know how any of these species are related, which makes me feel that your conclusion in Line 376-377 is overstepping the evidence presented. For example, the complicated patterns of color polymorphism, plant use and broad sympatry among Heliconius clades makes me cautious about jumping to such conclusions here. Personally I would like to see a detailed phylogeny of all populations and closely related species before I drew this strong of a conclusion.

Minor points –
I was not familiar with the term “gentian” and was unsure if it was a typo or not when I first read it, so I would suggest that that you define it the first time that you use it.
4 lines of refs (78-81) seems a little excessive
Line 125 – we can only get 95% so would have to make 96 from pure ethanol. Is this a typo, or do you dilute ethanol to make your preservative?
Line 137 – which unpublished data are you referring to here? I am not sure I follow this sentence.
Line 153 – there is not enough information for someone to recreate your PCR from this. Please provide the full details so that the reader could repeat your method.
Line 157 – how did you determine the proportion of missing alleles? Is this your nulls? You used Micro-Checker (lines 164-166), so is that the number you are reporting here, or is this the loci that failed to amplify in your experiment? I am not sure what you are trying to explain to us here…
Line 163 - this is a very low number of permutations for these tests. I don't know if it will change any results, but this is not up to the standards of the literature.
Line 229 – run-on sentence
Line 294 – M. alcon should be italicized. There are a few examples of this (also see line 365), so please check through and correct the plain text species names.
Line 302 – “this study [confirms]…”
Line 312-315 – I am not sure what you mean in this sentence. Please revise for clarity
Line 317 – no gene flow seems a pretty strong conclusion with such low FST values and on the order of 5% genetic variation distributed among populations.
Line 322 – “higher within-nest than between-nest relatedness [among] individuals…” and this statement raises the question for me as to why there isn’t high inbreeding coefficients? Shouldn’t this scenario you describe lead to substantial inbreeding?
Lines 330-334 – you say that the forms are more genetically similar locally than either is to more distant sites with the same host plant use, but that conclusion seems at odds to me if R-dry is more similar to Sardu than to R-wet? Either there is something wrong with this sentence or you totally lost me here...
Line 340 – The data could also support the exact opposite conclusion with a diverse ancestor and a recent spread of a homogeneous wet clade. You really need a good time-constrained phylogeny to infer the age of the groups, so this is speculation, but I cannot tell if it is your hypothesis or theirs here. I am fine with you posing hypotheses, but they should really be worded as such instead of cast as conclusions when you lack the data to test this...
Line 345 – I would say that it’s “little”, not "no evidence" - ant usage explains 2% of your variation in the AMOVA, even if it is not significant…
Line 346 – you don’t provide the reader with the data on genetic diversity, so it is hard to evaluate this statement. I would like to see what statistic you used to measure genetic diversity and the value for it across populations to evaluate this…
Line 347 - You give us GPS but not a map or actual distance in the paper - we don't know your study system, so please tell us exactly what "relatively large" is, and what "potential barriers" exist...
Lines 351-369 – what about other systems or other species? Seems like you could compare how your findings are different from, say the Heliconius system, here and put your work into a broader context for the reader. Similar differentiation among host plants has been demonstrated for a range of other insect species, and even among congeneric sea slugs specializing on different host corals – Faucci, A., Toonen, R. J., & Hadfield, M. G. (2007). Host shift and speciation in a coral-feeding nudibranch. Proceedings of the Royal Society of London B: Biological Sciences, 274:111-119. How does your system compare and support generalizations, or why is it different?
Lines 353-355 – this seems like an argument for the Orlog model of conservation - Bowen, B. W., & Roman, J. (2005). Gaia's handmaidens: the Orlog model for conservation biology. Conservation Biology, 19:1037-1043.
Line 358 – “This is” - unclear antecedent
Line 376 – unclear antecedent
Table 2 – what is “No. genetic samples” here exactly? Is this your sample size? If so, your legend should explain this, and it makes me more interested to see the AMOVA and pairwise tables because I am not comfortable with making those comparisons with a sample size of 1 or 2 individuals, so I need to see exactly how your samples were pooled (or not), and how these samples were user in your analyses… Based on your Structure plots it looks like all samples were pooled (i.e., Sardu sample size is 13 individuals), but it is not clear from the text if that is also true in the AMOVA analyses…
Figure 2 – The colors on the legend do not come through well on my screen and I can't tell which is blue, orange or purple based on the tiny dot – please increase icon size to make sure the color is visible on the final figure.

·

Basic reporting

The topic of the paper is interesting especially for people that work on butterfly conservation since the lack of evidence of a separation between the two taxa is the cause of a lack of conservation of the two ecotype.
For this reason many authors try to find out experimental evidence of two taxa. All molecular approach failed and the ecological evidence remain the only ones. In this respect author provide a very well written introduction to the problem but I’d suggest to put better in evidence the difference between previous approach and the method they used.

In general the manuscript is clear but I think is too long since probably the English could be meliorate but I’m not the right person to say that. For example in abstract lines 28 and 29 is written twice the verb using .
Figures and tables are adequate except for table 2 where all numbers are different from the text please check carefully.
Line 281 PCA explain 52% of variability but the first two principal coordinate only the 23.7 according to fig 2 so probably some other principal coordinates explain a weighty part of variability. In this case it would be interesting see the output of the analysis.

Experimental design

the experimìental design is appropriate. Authors know well the approach to use to investigate this problem
Methods are really detailed but may be they can be move in a additional material section

Validity of the findings

Findings are let me say unfortunately in perfect accordance with all previous studies and make the conservation of this species not easy. On the other hand authors produce for the first time population structure of two sympatric populations of the two M. alcon ecotype and they stress the role of the food plant phenology. That is interesting after years of studies on the role of the host ant and is in my opinion useful also to explain other similar situation like Thymus or Oreganum M. arion populations.

Additional comments

I'd mention the fact that hygrophilous alcon populations are severely threatened in a lot of countries while the xerophilous form is in better situation but the Habitat Directive considering all population the same species do not produce any legal tool to save them.

---

## Round 0.2 · accepted · Accept

Thank you for your detailed responses and the careful revisions that you have completed on your manuscript. I am satisfied with your responses to all the comments from the previous round of review, and hope you agree with me that the manuscript has been substantially improved through revision. At this point, I am happy to move your manuscript forward in the process, and look forward to seeing the final product.